

# How to be a great dad: parental care in a flock of greater flamingo (*Phoenicopterus roseus*)

Camillo Sandri[1], Vittoria Vallarin[2], Carolina Sammarini[1], Barbara Regaiolli[3], Alessandra Piccirillo[4] and Caterina Spiezio[3]

[1] Department of Animal Health Care and Management, Parco Natura Viva - Garda Zoological Park, Verona, Italy
[2] Dipartimento di Scienze Chimiche, della Vita e della Sostenibilità Ambientale, University of Parma, Parma, Italy
[3] Research and Conservation Department, Parco Natura Viva - Garda Zoological Park, Verona, Italy
[4] Department of Comparative Biomedicine and Food Science (BCA), University of Padua, Padua, Italy

## ABSTRACT

In the last years, studies on captive greater flamingos have increased. Research on zoo animals is important to improve the knowledge on these species and to improve their *ex-situ* and *in-situ* conservation. The aim of the present study was to investigate the parental behaviour of a captive colony of greater flamingo hosted at Parco Natura Viva, an Italian zoological garden, to improve the knowledge on this species in zoos. In particular, the present study investigated and compared the parental care of females and males in 35 breeding pairs of greater flamingos. For each pair, we collected durations of parental care behaviour of both females and males, recording their position in relation to the nest (near the nest, on the nest, away from the nest) and individual and social behaviours performed. First, both partners were involved in parental care and displayed species-specific behaviours reported in the wild. The main results were that males spent more time than females on the nest ($P = 0.010$) and near it ($P = 0.0001$) and were more aggressive toward other flamingos than females, both when sitting on the nest ($P = 0.003$) and when near the nest ($P = 0.0003$). Therefore, male flamingos seem to be more involved in incubation duties and nest protection than females. This kind of research is important not only to expand the knowledge on bird species such as flamingos, but also to improve their husbandry and breeding in controlled environment. Indeed, understanding animal behaviour allows us to gain insights into their individual and social needs, addressing potential animal welfare issues.

Corresponding author
Barbara Regaiolli,
barbara.regaiolli@parconaturaviva.it

## INTRODUCTION

Studies on wild flamingos are challenging to carry out. Indeed, flamingos are highly gregarious birds that live and breed in large dense flocks often including thousands of pairs (*Pickering, Creighton & Stevens-Wood, 1992*; *Johnson & Cézilly, 2009*). Obtaining information and data on their behaviour in the wild is therefore difficult due to constraints such as individual identification and approach to the birds (*Studer-Tiersch, 1975*; *Studer-Thiersch, 2000*; *King, 2000*). Zoo environments provide researchers with relative accessibility
to the animals and are important to obtain data on exotic animals, providing useful insights into the biology of their wild counterparts (*Hosey, Melfi & Pankhurst, 2013*). Despite research in zoological gardens might be compromised by confounding variables such as reduced sample size and human presence, the zoo environment houses animals in naturalistic group and enclosures, allowing the collection of data having a great biological relevance (*Hosey, Melfi & Pankhurst, 2013*). For these reasons, together with long-term studies on wild flamingo flocks, research on zoo colonies might be valuable and complementary to improve the knowledge on the ethology, morphology, physiology and endocrinology of these species (*King, 2000*). Moreover, studying the behaviour of flamingos in both wild and captive settings is important not only to improve our knowledge on the species biology but also for the implementation of their husbandry and their breeding (*King, 2000*; *Melfi, 2009*; *Rose, Croft & Lee, 2014*). Indeed, understanding species-specific behaviours, their function and causes, would be useful to know the animal feelings, preferences and needs, detecting and addressing potential welfare issues (*Mench, 1998*; *Hosey, Melfi & Pankhurst, 2013*).

Greater flamingos display sexual body size dimorphism as males are larger than females (*Johnson & Cézilly, 2009*). Greater flamingos are serially monogamous birds but they can form long-term pair bonds, at least in a zoo environment (*Pickering, 1992*; *Bagemihl, 1999*; *Johnson & Cézilly, 2009*; *Rose & Croft, 2015*). Both partners work together to build a nest, in which the female lays a single egg. The nest is usually a mound made of a mixture of mud/soil and sand, with a concave centre. It is generally built on an island or on the coastline of a lake (*Studer-Tiersch, 1975*; *Beletsky, 2006*; *Cézilly, 1993*; *Elphick, 2014*). Before egg laying, the male is primarily involved in nest building, but the female takes over as the laying time approaches. The nest building activity of both partners proceeds also during the first two weeks of incubation, leading to an increase in the nest height (*Studer-Tiersch, 1975*).

After copulatory behaviour, the female lays one egg in the nest. Then, both females and males take part in the incubation, lasting from 27 to 31 days (*Beletsky, 2006*; *Cézilly, 1993*; *Elphick, 2014*). Male greater flamingos have been found to make a greater effort in incubation (*Rendón-Martos et al., 2000*). When incubating the egg, flamingos display different behaviours, such as standing, stretching the wings, preening, self-scratching and looking at the nest (*Studer-Tiersch, 1975*). In addition, they take care of the egg, moving it with the bill. Flamingos could either stand or sit on the nest (*Studer-Tiersch, 1975*). While nesting, individuals of a pair including both the incubating bird and the non-incubating bird may perform aggressive display behaviour (*Studer-Tiersch, 1975*).

The aim of the present study was to investigate the parental behaviour of a captive colony of greater flamingo hosted at Parco Natura Viva, an Italian zoological garden to improve the knowledge on this species in zoos. In particular, the present study investigated and compared the parental care behaviour of females and males in a flock of greater flamingos, to verify the presence of both qualitative and quantitative differences between the study birds and their wild counterparts. The results of the study are discussed with the behavioural pattern shown by greater flamingos in the wild, to suggest strategies to improve the husbandry of this species in captivity and safeguard its welfare.
## MATERIALS AND METHODS

### Study subjects and area

The study was carried out in a flock of 147 greater flamingos of different ages (from 1 to over 20 years), 70 females and 77 males, housed at Parco Natura Viva - Garda Zoological Park in Italy, in a 1,100 m² open-topped enclosure. The study subjects were 35 heterosexual breeding pairs (approximately 50% of the study flock), during the peak of their breeding activity. The colony was composed of several juvenile immature individuals that were not involved in pair formation. Same-sex pairs and trios incubating an egg were not involved in the current study.

The enclosure was composed of a muddy area and a grassy area. The muddy area surrounded a water pool with two islands, used by flamingos to build their nest mounds and rear the chicks. The basal structure of the nest mound was built by humans, whereas flamingo pairs completed the nest construction properly, using mud, soil and sand naturally present in the enclosure. The nests had a density of approximately 2.5 nests/m². In particular, the nesting island had an area with an higher density of nests while others had a lower density. Trees, bushes and rocks were present in the enclosure, together with a wooden house to provide the flamingos with protection from weather conditions and a long feeding station. To minimize human disturbance, food was administered to the flamingos once a day in the feeding point. No interactions between humans and flamingos were allowed. The flamingo diet was composed of a specific pellet, containing cereals, vegetables, oils and fats, algae, shellfish, vitamins and mineral salts.

Flamingos were identified through a ring on one leg. The ring differed in colour and letters (three-letter combination). At the time of the study, the density of the flamingos in the enclosure was 0.13 individuals/m². Subjects of the study were pairs that incubated an egg in the 2016 breeding season ($N = 35$). To avoid excessive inbreeding, juvenile individuals were transferred regularly (approximately every five to six years) to new flocks in other zoological institutions. However, no bird transfers from and to other zoological gardens were done during the study period and in adjacent breeding seasons and the flock increased only due to flamingo reproductive activity. The study flock size ranged from 88 greater flamingos in 2012 to 147 in 2016, the year of the current study (*Sandri et al., in press*).

The study was carried out using non-invasive techniques, through live observation of the birds. The research procedure was in accordance with the EU Directive 2010/63/EU for animal research.

### Procedure and data collection

Subjects of the study were heterosexual breeding pairs and data collection started at the moment when a female laid an egg. For each pair, a total of twenty 10-minute sessions were carried out during the incubation period. In particular, for each pair two sessions per day were done, one in the morning (between 9.00 and 12.00) and one in the afternoon (between 14.00 and 17.00). To have data distributed over the entire study period, the observation sessions were carried out randomly within the incubation period. Data were collected using a continuous focal animal sampling method to record durations of behaviours performed

**Table 1  Behavioural ethogram of the study.** For each behavioural category collected in the study the table reports the name and definition.

| Behavioural categories | |
|---|---|
| Agonistic behaviour | During nesting, a flamingo extends the neck and peck at another bird or performs threatening displays, particularly neck hooking and swaying (*Studer-Thiersch, 2000*; *Johnson & Cézilly, 2009*) and feather spreading (*Schmitz & Baldassarre, 1992*). Agonistic behaviour can be performed when the flamingo is near the nest and on the nest (both standing and sitting). |
| Incubation | A flamingo is incubating the egg, keeping it warm with the body heat. This behaviour is performed when the flamingo is sitting on the nest. |
| Comfort behaviour | A flamingo is self-preening (trimming or dressing the own feathers with the beak), stretching (drawing out/extending legs or wings) or scratching the neck with one leg (*Studer-Tiersch, 1975*; *Brown & King, 2005*). Comfort behaviour can be performed when the flamingo is near the nest and on the nest (both standing and sitting). |
| Egg care | A flamingo is looking at the egg or rolling and moving the egg carefully (*Studer-Tiersch, 1975*; *Brown & King, 2005*), to assure even heating or cooling and prevent embryonic malformation (*Hauber, 2014*). Egg care can be performed when the flamingo is standing on the nest. |
| Nest-building | A flamingo looks for mud, soil and sand and picks them up to renovate or repair the nest (*Studer-Tiersch, 1975*; *Brown & King, 2005*). Nest-building can be performed when the flamingo is sitting on the nest. |
| Sleeping | A flamingo rests with the head under the wing. Sleeping can be performed when the flamingo is near the nest or sitting on the nest. |
| Other | A flamingo performs a behaviour that is not directly associated with parental care and is not listed above. This behavioural category can be performed when the flamingo is near the nest. |

by each flamingo partner (*Altmann, 1974*). One observer (V.V.) collected the data on all the study breeding pairs. Before commencing systematically data collection, the observer spent one month observing the flamingo behaviour whilst the birds get used to her. In addition, she carried out preliminary observations (40 h) in order to establish the data collection methods and to prepare an ethogram to be used to collect behavioural data. The ethogram (Table 1) was made basing on the studies of *Studer-Tiersch (1975)*, *Schmitz & Baldassarre (1992)*, *Brown & King (2005)*, *Johnson & Cézilly (2009)*, *Hauber (2014)* and was adjusted to this study basing on the preliminary observations.

Data collection was carried out following *Altmann (1974)*, using the focal sub-group sampling method in the presence of an egg on the nest. The chosen sub-group was the pair and the behaviour of each partner of the pair was collected during the session. For each pair, we conducted observations of parental care behaviour of both female and male, recording the position of the bird in relation to the nest and the behavioural category
performed. During each session continuous observation of each partner was done when flamingos were on the nest or near the nest. We collected durations of individual and social behaviours performed by male and female partners. In addition, to be accurate, when one partner was away from the nest and not under observation, the duration of "away from the nest" was also collected. In other words, we did not collect the behaviour of the subject away from the focal point as "under most circumstances, the only condition under which such a record can be obtained is that in which all the individuals in the sample group are continuously visible throughout the sample period" (*Altmann, 1974*).

In particular, regarding the position of the bird, we recorded the time spent by each flamingo partner near the nest (less than 150 cm, which is approximately the higher flamingo body length; *Del Hoyo, Elliott & Sargatal, 1992*), on the nest or away from the nest (>150 cm).

To consider the diurnal variability in the behaviour of the pairs, focal-sampling pairs were observed in a variable order. The focal-sampling of each pair started at precise moment of the morning and of the afternoon, following a regular schedule. Over the data collection sessions, such schedule allowed us to collect the behaviour of the bird in the whole time period from 9.00 to 12.00 and from 14.00 to 17.00.

As the observed behavioural categories were mutually exclusive, not overlapping, during focal-pair sampling we recorded the transition times, meaning that a behaviour ended when a new behaviour started.

### Statistical analysis

Kolmogorov–Smirnov goodness-of-fit tests revealed that not all data were normally distributed. Therefore, non-parametric statistic tests were used. In particular, Mann–Whitney tests were run to compare the duration of both positions and behaviours between females and males.

## RESULTS

### Reproduction of the greater flamingo flock in 2016

First, to be able to interpret more in deep the results on greater flamingo parental care behaviour we provide some data on the reproduction of the study flock in 2016 breeding season. In the current study, 27 out of 35 breeding pairs had a chick (77%). Moreover, within the 27 successful pairs, only two of them (7.4%) had to lay more than once before obtaining a successful hatch.

### Position of female and male flamingos in relation to the nest

Among female and male flamingos, significant differences were found in the time spent in different positions relatively to the nest. Females spent on average 8.7% of the observed time near the nest, 44.7% on the nest and 46.6% away from the nest whereas males spent on average 25.7% of the observed time near the nest, 55.3% on the nest and 19% away from the nest (see Figs. 1, 2 and 3 for medians and IQR). Mann–Whitney tests revealed that males were near the nest (Fig. 1) and on the nest (Fig. 2) significantly more than females ($U = 140$, $P < 0.00001$, and $U = 393$, $P = 0.010$, $N_1 = N_2 = 35$, respectively). On

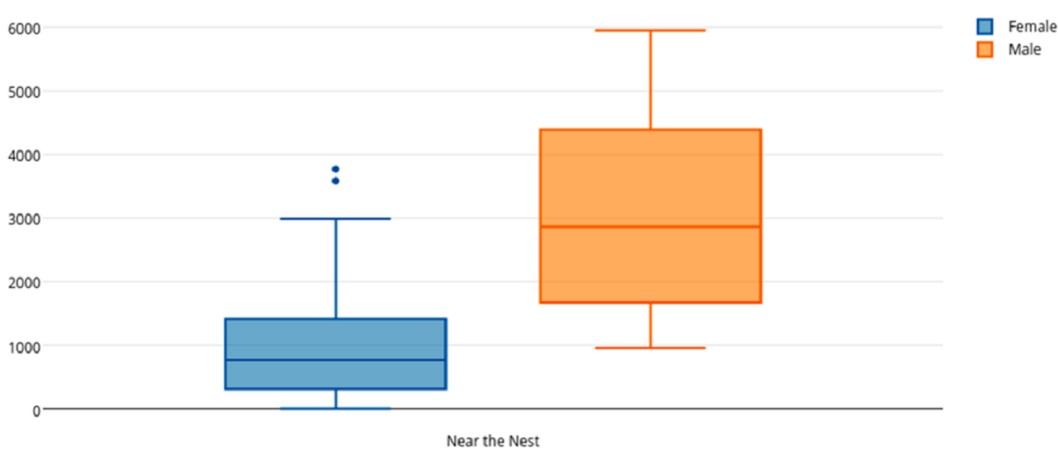

**Figure 1 Box and whisker plot of the time spent (seconds) by flamingo partners near the nest.** The horizontal lines within the box indicate the medians, boundaries of the box indicate the 25th and 75th percentile and the whiskers indicate the minimum and maximum values of the data samples. Outliers are drawn as points.

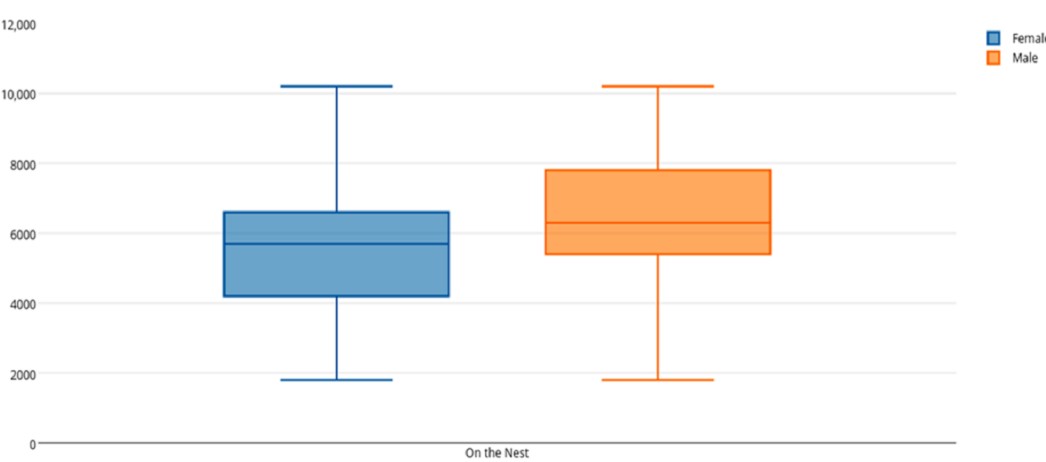

**Figure 2 Box and whisker plot of the time spent (seconds) by flamingo partners on the nest.** The horizontal lines within the box indicate the medians, boundaries of the box indicate the 25th and 75th percentile and the whiskers indicate the minimum and maximum values of the data samples. Outliers are drawn as points.

the contrary, males were away from the nest significantly less than females (Fig. 3) ($U = 121.5$, $P < 0.00001$, $N_1 = N_2 = 35$).

When flamingo partners were on the nest, we compared the time spent standing and sitting on the nest between female and male flamingos. When on the nest, females spent on average 5.5% of their time standing and 94.5% sitting whereas males spent on average 3.9% of their time standing and 96.1% sitting (see Table 2 for medians and IQR). Mann–Whitney tests revealed that males spent significantly more time than females sitting on the nest ($U = 375$, $P = 0.005$, $N_1 = N_2 = 35$), whereas no significant differences were found in the time spent standing on the nest ($U = 602$, $P = 0.905$).
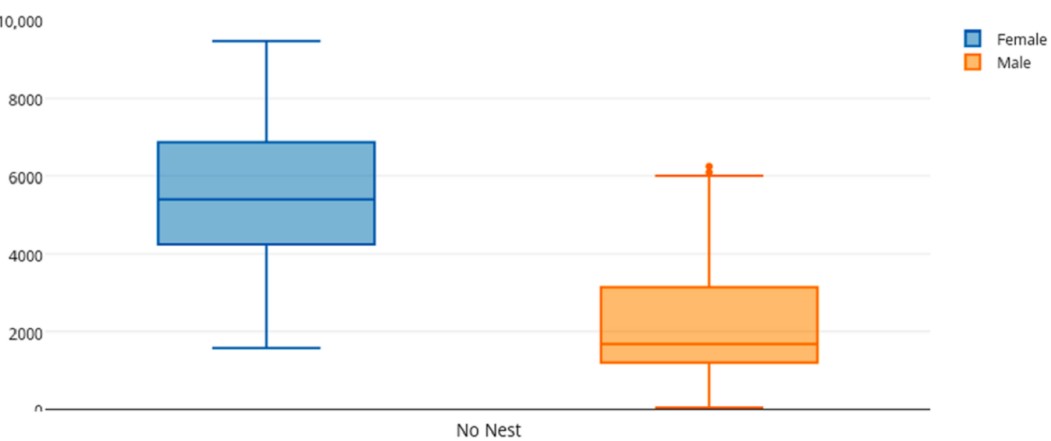

**Figure 3 Box and whisker plot of the time spent (seconds) by flamingo partners away from the nest (No Nest).** The horizontal lines within the box indicate the medians, boundaries of the box indicate the 25th and 75th percentile and the whiskers indicate the minimum and maximum values of the data samples. Outliers are drawn as points.

**Table 2 Behavioural categories performed by flamingos near the nest and on the nest (standing and sitting).** The table reports the median (IQR) duration in seconds of each behavioural category performed by females (F) and males (M) when they were near the nest, standing on the nest or sitting on the nest, incubating the egg. The last row reports the median (IQR) duration in seconds of time spent by female and male flamingos in different position.

|  | Near the nest | | On the nest (standing) | | On the nest (sitting) | |
|---|---|---|---|---|---|---|
|  | **F** | **M** | **F** | **M** | **F** | **M** |
| Agonistic behaviour | 40 (3–105.5) | 187 (40–326) | 11 (0–21) | 8 (0–19) | 545 (375–884) | 921 (637–1105.5) |
| Comfort behaviour | 231 (27–480) | 524 (210–945) | 14 (0–71.5) | 0 (0–24) | 64 (0–165.5) | 59 (4–230.5) |
| Sleeping | 55 (0–434) | 934 (358–1378) | – | – | 67 (0–634) | 319 (0–714.5) |
| Egg care | – | – | 148 (72–239.5) | 172 (99–320) | – | – |
| Incubation | – | – | – | – | 1,650 (1,081–1,895) | 1,995 (1,181–2,578.5) |
| Nest-building | – | – | – | – | 2,336 (1,523–2,956) | 2,791 (2,036–3,469) |
| Other | 255 (71–502) | 1,093 (432–1,836.5) | – | – | – | – |
| **Position** | **763 (287–1,405.5)** | **2,862 (1,654–4,365.5)** | **168 (114–380)** | **228 (99–385)** | **5,464 (4,010–6,067.5)** | **6,000 (5,238–7,248)** |

## Female and male activity near the nest and on the nest

When flamingo partners were near the nest, the behavioural categories observed were agonistic behaviour, self-directed comfort behaviour, sleeping and other activities not directly associated with parental care ("other") (Table 2). Mann–Whitney tests revealed that when near the nest males spent significantly more time than females performing the behavioural categories mentioned above: agonistic behaviour ($U = 300.5$, $P = 0.0003$, $N_1 = N_2 = 35$), self-directed comfort behaviour ($U = 319.5$, $P = 0.0006$, $N_1 = N_2 = 35$), sleeping ($U = 229$, $P < 0.0001$, $N_1 = N_2 = 35$) and "other" ($U = 198.5$, $P < 0.0001$, $N_1 = N_2 = 35$).

When flamingo partners were standing on the nest, the behavioural categories that we observed were agonistic behaviour, egg-care related behaviour (egg-care) and self-directed comfort behaviour (Table 2). Mann–Whitney tests revealed that males spent significantly less time than females in self-directed comfort behaviour ($U = 416.5$, $P = 0.021$, $N_1 = N_2 = 35$), whereas no significant differences were found for agonistic behaviour ($U = 600$, $P = 0.889$, $N_1 = N_2 = 35$) and egg-care ($U = 515$, $P = 0.254$, $N_1 = N_2 = 35$).

When flamingo partners were sitting on the nest, the behavioural categories observed were agonistic behaviour, attentive behaviour, nest-building, self-directed comfort behaviour (preening) and sleeping (Table 2). Mann–Whitney tests revealed that males spent significantly more time than females in agonistic behaviour ($U = 358.5$, $P = 0.003$, $N_1 = N_2 = 35$), whereas no significant differences were found for attentive behaviour ($U = 460$, $P = 0.073$), nest-building ($U = 474$, $P = 0.105$, $N_1 = N_2 = 35$), self-directed comfort behaviour ($U = 571$, $P = 0.631$, $N_1 = N_2 = 35$) and sleeping ($U = 551.5$, $P = 0.477$, $N_1 = N_2 = 35$).

## DISCUSSION & CONCLUSION

This study aimed at investigating the parental care behaviour of female and male greater flamingos, focusing on the time spent by each partner in different positions in relation to the nest and performing different behaviours. First, our results suggested that male flamingos spent more time on the nest and near it than females, whereas females stayed away from the nest (>150 cm) longer than males. The second main finding of this study was that male flamingos, when near the nest and when sitting on the nest, performed more aggressive behaviour toward conspecifics approaching the nest than females, protecting more intensively the egg and the nesting site.

Research on flamingo breeding behaviour is needed to improve the knowledge on these species, to find strategies that can improve their husbandry in a zoo environment and safeguard their welfare (*Ogilvie & Ogilvie, 1986*; *Hosey, Melfi & Pankhurst, 2013*). The aim of this study was to investigate the parental behaviour of a captive colony of greater flamingo to improve the knowledge on this species in zoological gardens. Firstly, greater flamingos of this study were found to perform species-specific behavioural repertoire (*Brown & King, 2005*). In particular, both flamingo partners were involved in parental care and displayed all the activities reported in the wild during incubation, such as moving and rotating the egg, nest-building, self-preening and stretching, nest protection and resting (*Studer-Tiersch, 1975*; *Pickering, Creighton & Stevens-Wood, 1992*; *Brown & King, 2005*; *Beletsky, 2006*; *Elphick, 2014*).

Results from the current study highlighted differences in parental care behaviour between female and male greater flamingos. Firstly, male flamingos of a breeding pair spent significantly more time on the nest and near it than females. These finding are in agreement with previous studies reporting a greater effort of male greater flamingos in incubation in wild settings (*Rendón-Martos et al., 2000*; *Rendón et al., 2014*). On the contrary, females remained away from the nest, without caring about the egg and the nest, longer than males. On the basis of previous studies, male flamingos take care of the

egg but do not feed their partner during the incubation process. Moreover, at least in the early stages, the parental investment is greater for females than males, due to the costs of egg-laying (*Cézilly, 1993*; *Johnson & Cézilly, 2009*). Although we did not collect data on flamingo behaviour away from the nest, it might be possible that since females remained less time on the nest and in proximity of the nest they spent more time looking for food, to recover from the egg-laying effort and replenish their reserves. Similar findings have been found in studies on breeding biology and parental care in shorebirds (*Lenington, 1984*; *Reynolds & Szèkely, 1997*). Further studies focusing on the behaviours performed by each partner away from the nest are needed to add data supporting this hypothesis.

When flamingos were on the nest, they could either be standing or sitting to incubate the egg. Our findings suggest that male flamingos spend more time sitting on the nest, thus incubating the egg than females, although no differences between sexes for standing on the nest were reported. Together with previous results on nest attendance (being near the nest or on the nest), these findings suggest a greater involvement of male flamingos in the incubation process as previously reported in wild settings (*Rendón-Martos et al., 2000*; *Rendón et al., 2014*).

When flamingos were near the nest, males were significantly more aggressive at defending the nest from other individuals than females and performed more self-comfort behaviour, sleeping and other behavioural categories. It might be possible that, since males were more involved in nest defence, they remained near the nest for a longer time when their partner was on the nest, instead of going away similarly to females.

When flamingos were standing on the nest the most important behaviour was caring for the egg, moving or rotating it to improve the incubation effort. According to our results, females and males spent the same amount of time in the egg care, confirming previous findings on parental care in greater flamingos (*Studer-Tiersch, 1975*; *Elphick, 2014*) and Caribbean flamingos (*Brown, Shannon & Farnell, 1983*) in zoological gardens.

Finally, even when flamingos were sitting on the nest, incubating the egg, males spent significantly more time than females performing agonistic behaviour. Aggressive behaviour has been previously found to increase during the breeding season in flamingos (*Farrell, Barry & Marples, 2000*) and might be due to competition over mates and over nest sites and food resources, as well as for nest/chick protection (*Johnson & Cézilly, 2009*; *Hinton et al., 2013*). According to our results, both female and male flamingos displayed agonistic behaviour. However, male flamingos were more aggressive than females when they were either near the nest and on the nest, incubating the egg. The study greater flamingos performed aggressive behaviours such as pecking at other birds or threat displays such as neck hooking and swaying (*Schmitz & Baldassarre, 1992*; *Studer-Thiersch, 2000*; *Johnson & Cézilly, 2009*) mainly to keep other birds away from their nest. Thus, our findings suggest that male flamingos are also largely involved in nest/chick protection.

In the current study, we focused on the parental care behaviour of the partners after the egg was laid and the nest was almost completed. The lack of differences in nest building between sexes reported in the current study seems to confirm that during incubation, after the egg is laid, nest building duties are equally shared by both partners, as reported in previous research on zoo greater flamingos (*Studer-Tiersch, 1975*).

In wild greater flamingos, the hatchling success is about 30% due to a high frequency of predation and other threats, almost absent in captivity (*Brown & King, 2005*). In the current study, 77% of the breeding pairs had a chick. Moreover, although wild flamingos lay one egg per breeding season, if it fails to hatch or the chick dies in the early stages, another egg might be laid (*Pickering, 1992*). In our study, only two pairs (7.4%) had to lay more than once before obtaining a successful hatch. Taken together, these findings suggest that more than three quarters of the eggs hatched and relatively few flamingo pairs had to lay more eggs before having a chick. Parental care of the study colony seems therefore to be linked to a good pair productivity in terms of number of eggs laid, as only one deposition is generally sufficient to have a chick. Future studies should be done to better investigate the relationship between greater flamingo parental care behaviour and their reproductive success.

Studies on greater flamingo social biology and pair bonding are relevant to enhance our understanding of the ethology of this species (*Rose, Croft & Lee, 2014*). Our data on the parental care of female and male greater flamingos can be added to the previous literature on this species. This kind of research is important not only to expand the knowledge on bird species such as flamingos, but also to improve the situation of captive animals and have a greater scientific understanding of issues important to modern zoos and *ex-situ* conservation. Indeed, species-typical behaviours of the animals are strongly informative of their feelings, preferences and needs and might therefore provide useful information to detect and address welfare issues (e.g., abnormal behaviour) and to enhance their daily lives (*Mench, 1998*).

## ACKNOWLEDGEMENTS

We would like to thank Dr. Cesare Avesani Zaborra for allowing this study to take place in Parco Natura Viva. Furthermore, special thanks should be given to Giuseppe Alampi, Parco Natura Viva flamingo zoo-keepers, for his hard work and for their precious help with the flamingo census and monitoring. Finally, we thank Paul Rose, Sunny Nelson and Jean-Loup Rault for their careful review of our manuscript and the useful comments and suggestions.

### Funding
The authors received no funding for this work.

### Competing Interests
The authors declare there are no competing interests.

### Author Contributions
- Camillo Sandri conceived and designed the experiments, performed the experiments, analyzed the data, wrote the paper, prepared figures and/or tables, subject census.

- Vittoria Vallarin and Carolina Sammarini performed the experiments, analyzed the data, subject census.
- Barbara Regaiolli analyzed the data, wrote the paper, prepared figures and/or tables, reviewed drafts of the paper.
- Alessandra Piccirillo analyzed the data, wrote the paper, reviewed drafts of the paper.
- Caterina Spiezio conceived and designed the experiments, analyzed the data, wrote the paper, reviewed drafts of the paper.

## Animal Ethics

The following information was supplied relating to ethical approvals (i.e., approving body and any reference numbers):

This study was carried out using behavioural observations of the flamingos, complying with the European Directive 2010/63/EU for the use of animals in research. Data collection was carried out in conformity with husbandry routine of the hosting facility and in maximum respect of the animal welfare.

## Data Availability

The raw data has been supplied as a Supplementary File.

## Supplemental Information

Supplemental information for this article can be found online at http://dx.doi.org/10.7717/peerj.3404#supplemental-information.

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
