# Peer review of "How to be a great dad: parental care in a flock of greater flamingo (Phoenicopterus roseus)"

_PeerJ, doi:10.7717/peerj.3404_

## Round 0.1 · original submission · Major Revisions

Thank you for submitting your manuscript to PeerJ. After careful consideration, both from my own reading of your manuscript and reading the comments submitted by three reviewers, I feel that it has merit but does not meet PeerJ’s publication criteria as it currently stands.

All three reviewers noted that your claims of welfare evaluation were not supported by the methods you used and I agree with them on this point. Although you provided a detailed ethological study of the flamingos’ natural behaviors, you failed to show how this demonstrated good welfare. This is a key area that must be revised, and Reviewers 1 and 3 provide detailed advice on how to do so. Not only have you not demonstrated how this is an effective evaluation of welfare, but you have also not shown how the birds’ behavior relates to fitness. Specifically, by only reporting on the birds’ behavior during the incubation period, without providing data on how successful their strategy was (i.e. how many chicks survived), the data are not especially informative. Finally, by simply reporting on how this specific flock of birds behaved without providing an evaluation of an intervention these data have little applied value for use by other zoos. As Reviewer 1 notes, it might be worth determining whether you wish this to be a behavioral report or an applied report, rather than attempting to provide both. Please take the time to strengthen your reporting and the conclusions you draw from your results throughout. Additionally, and as noted by Reviewers 1 and 3, the description of your methods would be clarified by the provision of the ethogram that you used to record the flamingos’ behavior.

I have three additional specific comments that build on comments made by reviewers 1 and 3:

First, and expanding on a comment made by Reviewer 1, if you collected data using a continuous focal animal sampling method, please provide more detailed information about this. Specifically, did you record the birds’ position (on, near, or away from the nest) as (i) a state behavior, (ii) as a modifier to each behavior you recorded, or (iii) their position at the beginning of the focal follow? Please clarify this as it will affect the interpretation of your results (i.e. whether you’re comparing % time or point sample data).

Second, and as noted by Reviewer 3 regarding your results, in your discussion you speculate that the female flamingos, when away from the nest, might spend more time looking for food. Do you have this information from your behavioral observations to answer this question? i.e. did you record the flamingos behavior when they were away from the nest?

Third, instead of simply comparing male and female behavior patterns, did you consider analyzing each pair as a unit and see if there was variation across pairs in their success or tendency to show species typical behaviors?

·

Basic reporting

These points relate to the abstract
- Line 22. I think there needs some context here. In comparison to mammalian species, all other species are as studied in zoos. But in the last few years or so, flamingo papers have appeared more frequently.
- What data were collected on these behaviours? Can you just state this in your abstract (i.e. minutes, counts of events etc.)
- The results in the abstract might benefit from the inclusion of some output from any inferential analysis so show differences.
- Line 36. How is the link between parental care and welfare made? This seems to have been included without any substantiation.
- Line 37. Maybe make some suggestions of how such research meets this husbandry and management aims?

These points relate to the introduction and use of supporting information.

- Line 45. The greater flamingo is of least concern with an increasing population trend. It is not correct to say that zoo populations are directly linked to conservation. You should consider how transferable this type of study is to species of flamingo that are of conservation concern.
- Line 48. Is this a conservation article? Or this is a behaviour and welfare article? Perhaps remove the conservation element and focus on measurement of behaviour and how this explains behavioural ecology, and hence animal welfare. There is a better story to be told here rather than trying to make this about conservation, when it clearly is not.
- The comments on the value of flamingo flocks in zoos to our understanding of their behaviour and flock structure are very valuable, and more should be made of this in the introduction, to be evaluated in the discussion.
- Line 59. Can you add some context to this statement?
- Line 61. Again, as per my comment in the abstract, yes, in relation to mammalian species. But flamingo behaviour papers have markedly increased recently (see Rose & Croft, 2017, App. Animal. Behav. Science for more details).
- Line 63/64. This sexual dimorphism can be very striking. What do you mean by reduced?
- Line 64/65. Needs context and a reference. Some authors state that flamingos can form new bonds each year. Others suggest that captive birds maintain longer bonds. Literature on wild association patterns for other populations and other species unknown.
- Line 66. The nest is made of ground? How do you mean? Flamingo nests are made of a mixture mud/soil and sand.
- The paragraph starting line 63 needs to be better linked to the theme of the introduction overall. You should try to explain key behaviours of importance and how these link to welfare, at the start of your introduction, and then say what these behaviours are in flamingos, and hence the aims of your study are to…
- Paragraph starting line 72. There is a lot of superfluous and unneeded detail here. Can you edit this down and explain only behaviours that are relevant to what you are investigating? Think about the title of your work and how your explanations of the animal’s biology help you to support your aims and research question.

Experimental design

Your aims are useful and have relevance to our understanding of flamingo behaviour in the zoo.
- Line 83. This aim is good. But it is not really supported by your introduction, which needs a more fluent behaviour and welfare theme.
- Line 87 to end of paragraph. These are not really aims. They are descriptive and would be better placed in your methods.

These comments relate to your methodology.
- The method needs an ethogram, ideally as a table, that defines and describes each behaviour that was observed.
- Line 95. Maybe give the proportion of the flock that actually breeds. It seems like it is around 50% of the total number of flamingos?
- Line 109. 182/sq.km. and this for only one population of birds that are at a high-resource feeding area. Context here is needed and the result from the paper needs to be quoted correctly.
- Line 112. What times of the day?
- Line 114. Data are plural.
- What was recorded? Time? Counts? Explain please. You say that you used continuous focal sampling but does this results in your data being in minutes or seconds of the activities observed? And did you time each behaviour consecutively and write down start and end points?

Methods appear repeatable and you have attempted to show what types of data were aiming to collect, but you need to provide further explanation of what you did and how it was done.

Validity of the findings

These comments relate to your results.
- Table 1. What data are these? Please give units. It might also be better to provide standard error rather than standard deviation.
- Are you able to provide any graphical information to show the range of times for each bird that performed each behaviour? E.g. box plots that show the range, median, quartiles and outliers of the behaviours measured at an individual flamingo level?
- Line 137 onwards. Are these results not simply repeated in the table?
- P values need to be presented as the test statistic, then the degrees of freedom and then the P value.
- Figures 1 and 2 are useful. They also show the units for all data, which would be helpful in the text (as per my comments above). Again, standard error bars might be more useful than standard deviation.

I think the results section is useful. But there seems to be repetition of data between Table 1 and what values are presented in the text. It would be good to have this section restructured so that you explain illustrative data in the text without repeating values are presented elsewhere. You should also consider not only presenting averages (again, as per my comments above)?

These comments relate to your discussion and conclusions.

- Perhaps start the discussion with what you have found, rather than general flamingo information.
- Line 184. What is the measure of good reproductive success? You should consider fertility or number of chicks raised per year. And only 35 pairs of a flock of 177 seems low?
- Line 197. Check format of the sentence.
- Line 198. Supposition. How do you know this is happening?
- Line 199. Flamingos do not multiple clutch.
- Line 207. "at defending the nest…" would read better.
- You provide a lot of citations in your discussion but you should explain the findings of their papers. So rather than saying “our results show this, like these people have found” you should explain what the finding of the paper is and in what context.
- Line 224. How does it show this?
- Line 238. Flamingos have incredibly high sexual selection. Your comment here should be reconsidered. Please see Perrot et al. (2016). Sexual display complexity varies non-linearly with age and predicts breeding status in greater flamingos.
- Line 239. You have not explained what the assessment of good welfare is. You have measured an activity pattern but you have not linked it to a measurable way of determining bird welfare. For the whole paper, think about how your experiment is trying to determine welfare from what has been measured.

Your discussion needs to explain the links between your findings and welfare. At the present, the link is not clear. What have you shown that is useful to our understanding of flamingo behaviour and management? How does this help explain positive welfare? You should explain how your findings link to the creation of sustainable flamingo flocks in captivity, rather than conservation aims.

For your reference list, please check your reference format. There are inconsistencies at times. Check that all proper nouns have capital letters (e.g. Dublin Zoo), and that journal names also have capital letters (e.g. Zoo Biology).

Additional comments

This is a very useful paper that investigates a novel and relevant area of flamingo biology, with sound applications to the zoo. You should consider what your paper is aiming to achieve. It is not a conservation paper, it is a paper that is explaining how flamingos organise their lives in captivity. You are gathering evidence to help us understand how they respond to a captive environment and how, then, we can ensure they behave normally. This normal behaviour and performance of important behaviours helps the birds to experience good welfare. Think about re-writing the theme of your paper, having a stronger between your supporting literature to your aims, and how your results help meet your aims, and then what these results mean for i) what we know about flamingos, ii) how we keep them, and iii) how we can ensure they breed in the same manner as wild birds.

·

Basic reporting

No comment

Experimental design

I am concerned with the conclusion of breeding success being equated to positive welfare. The design of this study appears to evaluate parental investment during the incubation period. There aren't any references cited pertaining to how this means this species or any other is experiencing positive welfare based on parental investment. I think it would be a fine publication if the main aim of this study, as described, was not to assess welfare based on breeding success, but instead to only determine parental investment of Greater flamingos.

Validity of the findings

The results during the study period are clear and easy to follow. The statement regarding the increase in flock size should be qualified with a statement regarding imports and exports of the flock. An increase in population size does not always occur through breeding, so the statement regarding an increase in flock size may be misleading if the population was supplement through imports as well as births.

·

Basic reporting

L.49: replace “has for them” for “is”/“is often”/“is often considered”
L.50-52: The paper would be strengthened by expanding your justification of the link between the performance of natural behaviours and their animal welfare implications.
L.64-65: it would be good to add a reference about the monogamous behaviour of greater flamingos, maybe reusing a reference from L.67-68 relevant to this aspect.
L.80: can you replace the term “short”, which is relative, for a more quantitative description such as “ 2-3 min” or “a few minutes”.
L.80-82: could you add information about whether they show territorial behaviour toward neighbour? (you only mention “disturbing”, which is a rather vague term). Also, how far apart flamingos placed their nest, in colonies?
Table 1: typo on nest building that is misspelled.

Experimental design

L.83: you have very limited measures to assess welfare, especially as animal welfare implies a lot more than just the ability to perform natural behaviours (only one of the 3 main concepts of animal welfare science assessment, as you know). Your study had a much more ethological approach. I suggest to reword your aims to emphasise the study of their behaviour, and only restrict the mention of welfare in the implications of this work, in the discussion. For example, you could replace the “welfare” term on L.83 by “ethology”, and mention welfare at the end of this paragraph, such as L.90: “to suggest strategies to improve the husbandry of this species in captivity, and safeguard their welfare”; I would suggest a similar form to the statement you have Line 176-177 at the start of the discussion which is a nice summary of the implications of this type of research.
L.99: add a mention on whether any material was provided to the flamingos to build their nest?
L.108-109: is that density in the wild true at all time, or just during breeding season? That could make a difference, as I assume that the density at the zoo doesn’t change between breeding and non-breeding seasons?
L.109-110: was the criterion of “incubating an egg during 2016” the only inclusion criterion used to select the 35 breeding pairs? Were there any exclusion criteria, or pairs that were initially selected but then dropped from the study (for any reason, for instance accidental egg breakage, sickness or death of one member of the pair)? It would also be useful to report the breeding success rates in that park, as a reference which people could compare to. You have a mention L.185-186 but it doesn’t state the specific success rate.
L.112: “in which the female laid the egg”, in opposite to what? Could they incubate an egg from a different pair? If this is the case, it would be interesting to report how often this special case occurs in that flock. Alternatively, if this is to simply specify the difference in roles between the male and female partners, it’s probably obvious and the statement could be removed.
L.115: you mentioned the typical incubation period range is 27 to 32 days. How were these 10 days placed in the incubation period, from the time the egg was laid, mid-incubation or late incubation period? Did it vary between pairs observed?
L.116: how many observers conducted the behavioural observations? Did you check for inter- and intra-observer reliability?
L.123-128: did you use an ethogram previously used in other published studies, and if so it would be worth adding the reference (possibly Brown & King 2005?). However, it seems you rather developed this ethogram for your own study. If you are able to add another Table, I think it would be worthwhile adding table laying out your ethogram, with the behavioural categories, behavioural type, and the definition for each behaviour as columns.
L.123-124: “Including aggressive interactions, such as extending the neck and beak at another bird”. Since agonistic behaviour is one of your main finding, you should add more details in that statement about all types of behaviour that count as “aggressive interactions”.
L.138 and elsewhere: add the unit, so “1,049.86 ± 994.80 cm for females …”.
L.137: I am not sure about the rules of the journal, but I think you could state in the Methods section, at the end in the statistical analysis section: “All results are cited as mean ± SD unless otherwise noted”. That way you can remove all the mean ± SD from the results section, making it easier to read.
L.137-142 and L.147-150: just a suggestion, but I felt the time spend on, around or away from the nest would be easier to read if they were expressed as the percentage of time relative to total observation time, such as “Females spent X % of the observation time on the nest, X % near the nest and X % away from the nest whereas males spent …”, with the total rounding up to 100% of their time budget which corresponds to your 2 x 10 min x 10 days = 200 min of observations per pair. Furthermore, since these data already appear in Figures 1 and 2, I don’t think you need to cite it in the text as well.
L.180-181: you need to specify what you mean by “abnormal behaviour”, as the overall term is vague: it could be interpreted as a behaviour that flamingos don’t perform in the wild (as you may imply from this statement) but it could also be a behaviour that is part of their repertoire but that is expressed in extreme (high or low) amount relative to wild conditions. All of those are included in the term “abnormal”.
Table1: What was attentive behaviour? It’s not mentioned in the methods section.

Validity of the findings

L.158: which behaviour were females spending their time on then? If your ethogram was exhaustive, one would expect that females spent their time doing something else.
L.185-186: you should mention in the methods which methods are in place to prevent excessive inbreeding, as this could possibly affect the behaviour of that flock.
L.186-187: I disagree with this statement; ant at the very least you need to cite on what basis you can conclude that the flock has good welfare. Firstly, the only measure you made were behavioural observations, as the breeding rates are not specifically mentioned at present in the results, just briefly mentioned in the discussion. Furthermore, as you state in your introduction, animal welfare is more often focused on the individual rather than the group (which becomes a more important topic than welfare in the field of conservation); hence that you have good breeding rates in the flock doesn’t say much about the welfare of particular individuals within that flock. Furthermore, this is a conundrum that is classic in the farm livestock industry: poor welfare usually results in poor biologically functioning (reproductive success being one measure of biological functioning), but good reproductive success doesn’t not necessarily equate to good welfare. Secondly, this statement relates to the theory that the display of natural behaviour is a faithful reflection of welfare states. Similarly to the introduction, you need to expand on that to strengthen the argument, as the use of natural behaviour as a basis for animal welfare assessment is not necessarily as evident as one thinks. The alternative option is to remove the reference to welfare from the aims of your study, and just mentioned it in terms of the implications of a better understanding of the behavioural repertoire of flamingos kept in captivity.
Line 225-226: this should come earlier, and be mentioned in the methods (see previous comment above on which 10-day period you study out of the 30-day incubation period.
L.239: according to previous comment, remove welfare and replace for “behavioural repertoire”.

Additional comments

Overall, the paper is well-written, clear and the experimental design appropriate. I disagree with the quite direct statements on welfare status being derived simply from the behavioural repertoire shown. The welfare of flamingo is an implication of the current work, but the aim was a behavioural observation study, rather than an animal welfare assessment (see specific comments). Details need to be added, mainly to the methods (see specific comments).

---

## Round 0.2 · accepted · Accept

Two of the original three reviewers had a chance to review this revised version of your article and both are in agreement that you have made considerable changes to the article, improving its clarity and focus. (One reviewer notes that you did not reference Table 1 in the text of the article, but I did see that it was referenced so you do not need to make an amendment there.)

I agree with both reviewers and am happy to accept your article for publication in PeerJ. Thank you for taking the time to make these considerable changes.

·

Basic reporting

I am happy with the changes made. The paper now has a coherent story, and all of the comments made by the reviewers have been actioned. This is a useful and worthwhile piece of science.

Experimental design

The experimental design has more detail, and the methods are repeatable. The aims of the study are clearly spelling out and it is evident what type of data were collected and analysed.

Validity of the findings

Very useful, and interesting, the authors have explained and evaluated all key areas of their results and these are put into a wider context. I am happy with how this paper has been edited and re-written.

Additional comments

As my comments above. This is a much tighter, and much more relevant paper to our understanding of flamingo behaviour in captivity and how this can inform care and husbandry in the zoo.

·

Basic reporting

I can see Table 1 in the revised manuscript, but it does not appear to be referenced in the text. Please add a reference to Table 1 in the methods section when referring to behavioural observations, such as "(Table 1)".

Experimental design

ok

Validity of the findings

ok

Additional comments

The revisions made are satisfactory